# Causal reasoning without mechanism

**Selma Dündar-Coecke**[ID][1]*, **Gideon Goldin**[2], **Steven A. Sloman**[2]

**1** Quantinuum, QBA, Centre for Educational Neuroscience, London, United Kingdom, **2** Department of Cognitive, Linguistic, and Psychological Sciences, Brown University, Providence, Rhode Island, United States of America

* selma.coecke@gmail.com

## Abstract

Unobservable mechanisms that tie causes to their effects generate observable events. How can one make inferences about hidden causal structures? This paper introduces the domain-matching heuristic to explain how humans perform causal reasoning when lacking mechanistic knowledge. We posit that people reduce the otherwise vast space of possible causal relations by focusing only on the likeliest ones. When thinking about a cause, people tend to think about possible effects that participate in the same domain, and vice versa. To explore the specific domains that people use, we asked people to cluster artifacts. The analyses revealed three commonly employed mechanism domains: the mechanical, chemical, and electromagnetic. Using these domains, we tested the domain-matching heuristic by testing adults' and children's causal attribution, prediction, judgment, and subjective understanding. We found that people's responses conform with domain-matching. These results provide evidence for a heuristic that explains how people engage in causal reasoning without directly appealing to mechanistic or probabilistic knowledge.

## 1. Introduction

Even experienced cyclists cannot reliably draw a picture of the mechanism that makes a bicycle work [1]. Moreover, few people can explain how a ballpoint pen works; in fact, when they try, they discover they do not understand such artifacts as well as they thought they did [2]. Most of us live with the illusion that we have more causal knowledge than in fact we do [3].

It is unrealistic to expect people to remember all that we learn about how things work. Things have many layers of complexity, both because they have parts that themselves can be decomposed at multiple levels, and because they interact with so many other things (understanding ballpoint pens requires understanding writing). So how do people make inferences about what causes what if things are too complex for them to understand? A number of proposals have been offered. Einhorn and Hogarth [4] suggested that people rely on several "cues-to-causality," including covariation, temporal order, contiguity in time and space, and similarity of cause and effect. Evidence for some of these cues has accumulated (see e.g., Lagnado & Sloman [5] for temporal order and contiguity; LeBoeuf & Norton, [6], for similarity; Johnson & Keil [7] and Rottman & Hastie [8], for more recent reviews). However, such cues only provide pairwise information about variables. Causal mechanisms generally involve sets of

**Data Availability Statement:** The data underlying the results presented in the Supporting information files.

**Funding:** Steven Sloman was supported by a grant from the Thrive Center for Human Development. In

addition, this publication was made possible through a grant from the Varieties of Understanding Project at Fordham University and the John Templeton Foundation. The opinions expressed in this publication are those of the authors and do not necessarily reflect the views of the Varieties of Understanding Project, Fordham University, or the John Templeton Foundation.

**Competing interests:** The authors have declared that no competing interests exist.

variables working together in a highly structured way (like the pedals, gears, chain, wheels, and frame of a bicycle) and pairwise relations are not sufficient. Johnson and Keil propose a heuristic for capturing a certain kind of structure—the hierarchical structure of events—by positing a level-matching principle.

Our proposal appeals to similarity indirectly and offers a heuristic for making causal inferences that respects structural relations, not merely cause-effect pairs. Our proposal assumes that retaining abstraction information from a small set of categories of mechanism is cognitively feasible. Whenever we lack specific knowledge about a process, we can note the broader type of that process to bootstrap causal inference. We refer to such types as "domains." Our claim is not that mechanisms come in a fixed hierarchy of types, but that any appropriate abstraction process induces a category and therefore a domain. Thus, causal domains tend to reflect regularities in the world; and we are in fact encouraged to consider only the likeliest causal relations, effectively reducing an otherwise overwhelming search space.

We propose that humans tend to make inferences and ascribe causal structures based on the domain of the corresponding mechanism. Categorical domains help to identify types of entities [9, 10], but mechanism domains enable us to identify the kinds of parts and processes that are causally related and operate in similar ways. For instance, the knowledge that the mechanism for making calls on a cell phone is electromagnetic is enough to guide a swath of inductive inferences. The belief that a cellphone uses an electromagnetic mechanism suggests that its performance depends on whether the phone case is made of metal, but not its color. In most cases, although a detailed understanding of how a cell phone works is not necessary to use it, mechanism domain knowledge helps to identify the type and scope of relevant information. It makes many inferences feasible and economical.

We propose *the domain-matching heuristic*, hypothesizing that we are likelier to assume two events are causally related if they share the same mechanism domain. When we observe a cause that participates in the mechanical domain, we are more likely to infer a corresponding effect that also participates in the mechanical domain. If we observe an effect in the chemical domain, we will look to possible causes that also participate in the chemical domain. Our goal in this paper is to test this proposal and explore whether people's causal attributions fit mechanism domains when they link potential causes and effects. We also conjecture that "cross-domain" mechanisms might be relatively rare, rendering the domain-matching heuristic a useful guide most of the time.

## 1.1 Causal attribution in the absence of mechanism knowledge

Causal processes can be reduced to mechanisms, sequences of interconnected events that involve parts and processes [11, 12]. In psychology, the Piagetian framework recognized this long ago, proposing a mechanistic view for causal systems, where mechanism refers to, for example, how a bicycle works. Johnson and Ahn [13] define mechanisms as systems of visible and invisible characteristics interacting systematically, where the same effects are produced by the same causes. Park and Sloman [14] define them as a set of causes, enablers, disablers, and preventers that are involved in producing an effect, unfolding over time. Lombrozo [15; see also 16] highlights that mechanisms have a privileged relationship to explanations; they do not simply identify causes but illustrate how the cause brought on the effect.

Exposure to mechanisms is inevitable. Within months of taking our first steps, we start establishing some appreciation of how the world works without developing a deep understanding of operating mechanisms. Children learn to keep their ice-cream in the fridge on a sunny day. Cooks will use an oven to bake a cake. How exactly do people think they understand phenomena without knowing the details of causal interactions between functional parts?

We argue that mechanism domains offer a heuristic for a variety of causal reasoning tasks so that even though most of us do not know how a fridge or an oven work, we rely on our beliefs that are rooted in knowledge of mechanism domains. It is this ability that helps us to organize knowledge into content domains like biology or chemistry.

Findings showing young children's ability to distinguish animals from artifacts supports this view in the sense that children demonstrate distinct explanatory understandings of, for instance, biological (animals) and mechanical causal agents (machines or blocks). They seem to believe there are distinct mechanisms driving the causal relations in different domains [17], with multiple causal-explanatory construals for physical, biological, psychological, and chemical events. For instance, when asked, a preschooler could state that hammers break things, whereas water makes things wet, possibly by associating causes with effects in a domain-specific manner [18–21].

Comparing young children's and adults' responses over a series of five experiments, Shultz's [22] study also showed that, for instance, in the physical domain, familiarity with the objects in a question is not a strong indicator of mechanism level thinking. In one of the experiments, where children and adults were presented with sound, wind, and light transmissions in different procedures, participants' tendency to analyze causal mechanisms was not restricted to prior knowledge. Even young children knew that a spot of light was likelier to be due to a flashlight than a fan, and furthermore, their justifications more often relied on knowledge about the nature of the transfer (e.g., light spots are round) than past experience (e.g., it looks like my flashlight).

The relational complexities inherent in most causal mechanisms seem to drive people to develop beliefs about certain relational patterns, with some explanatory frameworks allowing them to make sense of properties and causal relations. Intuitive and lay theories conjecture that these beliefs are intrinsically limited, incomplete, or partial models for how things work [23, 24]. Although the majority of everyday explanations invoke cause-effect relations, most without requiring domain expertise (the wind blew the fence down), people often seem to determine appropriate relationships through a mental process producing subjective beliefs about reality.

Consider someone who believes they will get sick if they fail to use soap. According to Ahn and Kalish [9], this thought implies a belief in a mechanism, whether it involve viruses, miasma, or something else. Walsh and Sloman [25] highlight that most of the evidence on reasoning about causal relations supports mechanism-based theories (see also [26]). Most of us do not know how soap works, but we act by relying on some beliefs about an underlying process. Ahn and Kalish explain this as "people's beliefs about causal relations include. . . [among other things]. . . a set of more or less elaborated beliefs about the nature of that mechanism, described in theoretical terms." (p. 5). In our view, these beliefs are rooted in knowledge of mechanism domains.

Across a series of experiments, Rozenblit and Keil [2] asked people for self-understanding ratings for how a number of devices worked. People's ratings were lower after trying to explain how the device worked, after seeing an expert explanation, or after they were asked a key comprehension question about the system suggesting that people think they know how things work better than they actually do [see also [3]).

What allows people to address causal questions when their causal knowledge is so impoverished? We propose that a mechanism-domain-matching heuristic is one of the strategies humans employ in new situations to close the gap between, on the one hand, causal explanation and prediction, and on the other hand, prior knowledge and understanding. Understanding, explaining, and predicting are intimately related but also distinct competences [27], the differences between them giving cues as to how people can predict and explain a causal phenomenon without fully understanding it.

With respect to the categorization of knowledge, this kind of representation would constitute what Sloman, Lombrozo, and Malt [28] call an extra-strong ontology, wherein differences between any proposed domains are irreducible to other cognitive representations. In their analysis of domain-generality versus domain-specificity in higher-order cognition, they outline four more possible ontologies, each of which is decreasingly committed to a strong distinction amongst domains of knowledge (where no ontology is the 5th and alternate extreme). They argue for mild ontology, which they describe as follows: "Domain differences in categorization and inference are systematic, but not cognitively primitive. People tend to reason and classify phenomena using domain-general causal reasoning mechanisms.

To the extent that domains correspond to causal discontinuities in the world, systematic differences between domains may emerge, and domains thus serve as a useful shorthand for theorists to roughly classify different types of processing. However, in a given classification or inference, an object is processed the way it is by virtue of its causal history and other causal roles, which will correlate imperfectly with its domain" (p. 201). This implies that mechanism domains are not parts of a pre-specified ontology, but rather, they emerge from our observations of causal regularities and thus we take our hypothesis to be a form of "mild ontology" in the language of Sloman, Lombrozo, and Malt [28]. We propose that mechanism domains constitute the fundamental representations that allow us to generate causal models and explanations quickly and effortlessly.

## 1.2 Overview of the studies

The proposed hypothesis that 'mechanism domains facilitate search by focusing our attention on likely relations' was investigated in 5 studies. The first study documented whether people's sorting tendencies generated any clusters when they are asked to sort artifacts by common mechanisms. This study found three mechanism domains: mechanical, chemical, and electromagnetic. Studies 2 and 3 tested one implication of the hypothesis: that people should select a cause that matches the domain of the effect. With a larger number of items, studies 4 and 5 examined whether people chose within-domain causes over cross-domain causes. Two exploratory studies (2b and 4b) examined younger people's (aged 8 to 17) choices and judgments of the same material to analyze the factors influencing use of the domain-matching-heuristic among youngsters.

All studies were conducted online. Only volunteer children and adults participated. The sample sizes across the studies were based on a pilot work as well as the availability of volunteer participants. Tests for normality of data have been run for each condition of each experiment. Except for the first study, the data were normally distributed. There was a significant skew on scores for gender in Study 1. However, this study was exploratory -participants sorted a set of items and hierarchical clustering was used to examine their classifications without the need for parametric analyses. In Study 4, where 88 children were recruited, main effects of age were followed up with tests assessing differences between age groups, using Bonferroni corrections.

Ethical approval for this research has been obtained from Brown University Research Ethics Committee. None of the participants needed to sign a written consent. Regarding children's data, only those whose parents signed the consent form were included. Parents received the online web link asking them to review the questionnaires and the target of the study. Those allowed their children to access it included.

## 2. Study 1: Uncovering mechanism domains

Previous literature has highlighted differences in people's reasoning about natural kinds versus artifacts [29] and in general, in the phenomena associated with different areas of study [30].

The purpose of this study is to determine how people would sort randomly sampled items: whether they sort different set of real-world items in similar or dissimilar ways, that is, in such a manner that a hierarchical analysis could reveal the domains of mechanisms. To begin, we ran a study to enumerate the domains people use to think about common artifacts. We asked participants to sort items based on their perceived mechanisms. Two other groups of participants were asked to sort the same items based on either function or overall similarity.

## 2.1 Method

Sixty participants were recruited from Mechanical Turk. Nine were removed for not passing an attention check, and one was removed for not answering all items (final N = 50; 34 male, 16 female; Mage = 32.40 years, SDage = 12.93 years). The study required an average of 12 minutes to complete.

We used the 42 artifact-based stimuli from Study 1 in Rozenblit and Keil [2] as our test items, as this set was of reasonable size and already constructed independently. Items included: quartz watch, zipper, spray bottle, solid-fuel rocket, VCR, cellular phone, helicopter, etc. (see S1 Appendix for the list of items).

Participants were randomly presented with the 42 items on screen. Each of three groups was given a different sorting instruction. Participants were randomly presented with the items on screen. They were then asked to drag and drop the items into distinct groups based on either (i) how similar the items' functions are (i.e., what they did; function condition), (ii) how similar the items' mechanisms are (i.e., how they worked; mechanism condition), or (iii) how similar the items themselves are (general condition). Participants completed the task by dragging and dropping the items into on-screen bins.

## 2.2 Results & discussion

On average, participants sorted the items into 12 groups (Moverall = 11.88; SDoverall = 5.11; Mmechanism = 11.64, SDmechanism = 5.42; Mfunction = 14.06, SDfunction = 4.97; Mgeneral = 10.11, SDgeneral = 4.47). For each condition, a hierarchical clustering was used to explore the nature of people's groupings. To do this, a 42-item x 42-item distance matrix was computed in each condition by assigning to every cell the number of participants that sorted the intersecting pair of items into distinct groups. Items along the main diagonal were set to the minimum distance, 0, while the maximum value possible for any cell was the number of subjects in the condition (Nmechanism = 14, Nfunction = 17, Ngeneral = 19). An agglomerative hierarchical cluster tree was then built, taking the original distance matrix as input and computing linkages between clusters using their unweighted average distances. Cluster membership for each item was obtained by pruning the binary tree for three top-level branches, proposing that participants maintained three mechanism domains in their sorting as shown below.

In the mechanism condition, slicing the tree just past a distance of 12 revealed three clusters (see Fig 1). One cluster contained many items that are chemical in nature (e.g., greenhouse, solid-fuel rocket), while another cluster contained mostly mechanical devices (e.g., piano, helicopter). The last cluster contained almost only electrical items (e.g., VCR, cellular phone). This division of items captured the mechanism domains, as opposed to capturing other domains at a different level of abstraction.

In the function condition, three clusters emerged when cutting the tree at a distance of about 16 (see Fig 2). This clustering depicts different patterns. Here, items tend to group around a purpose rather than a process, such as travel (e.g., solid-fuel rocket grouped with helicopter here, but not in the mechanism condition).

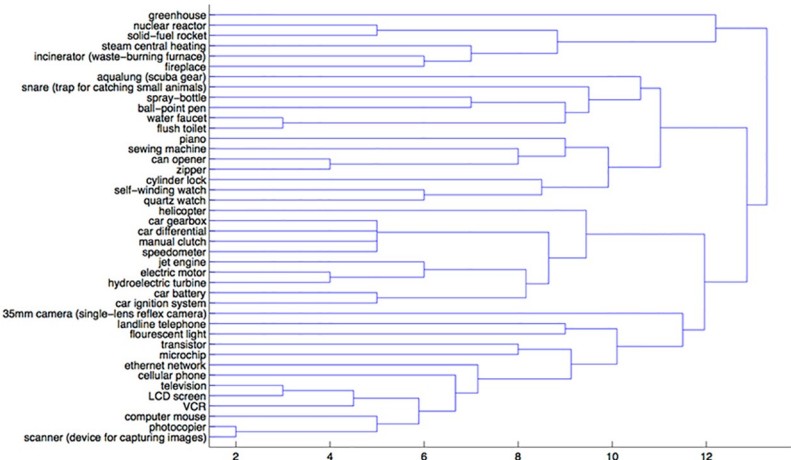

**Fig 1. A dendrogram representing the mechanism condition sortings.**

In the general sorting condition, electrical items tend to group together while the other two clusters were not as clear. It appears that participants considered function when judging similarity (see Fig 3). This behavior is supported by the HIPE theory of function [31, 32], whereby an object's physical structure is thought to be caused by its historically intended role (i.e., its function).

In order to compare clusterings, three metrics were employed, namely the Rand Index (RI: [33]), the adjusted Rand Index (ARI: [34]), and the Variance of Information criterion (VI; [35]). These metrics' results were shown in Table 1 below.

The RI is derived by counting the number of times a given pair of items are grouped together or separately in two clusterings divided by these counts as well as the number of times they are grouped inconsistently (i.e., together in one clustering but separately in the other). The index spans from 0 to 1, whereby identical clusterings obtain 1 and clusterings that share no consistency score 0. The ARI spans the same range but is designed to be more sensitive to differences of that are the result of chance. The VI takes an information-theoretic approach,

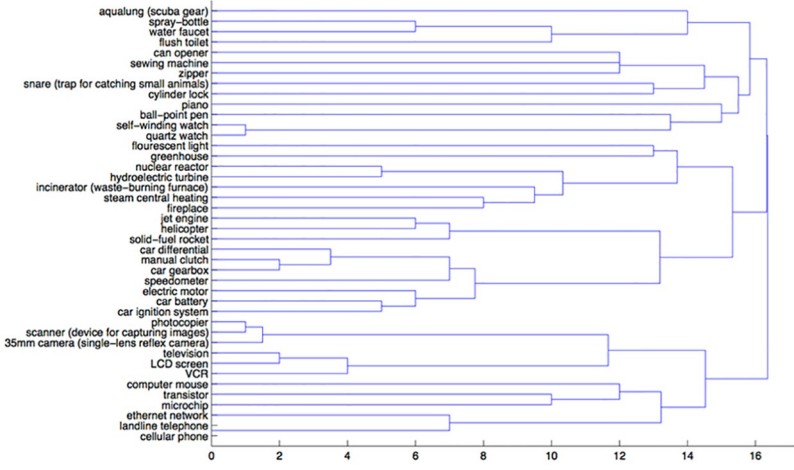

**Fig 2. A dendrogram representing the function condition sortings.**

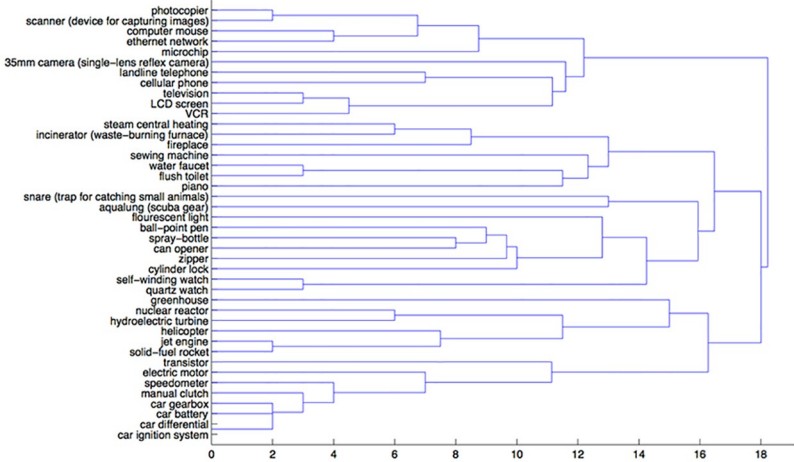

**Fig 3. A dendrogram representing the general condition sortings.**

attempting to discern how much information one clustering can provide about the other. As such, two identical clusterings would obtain a VI score of 0 (i.e., no information can be gleaned about the other), and the VI has an upper bound of log $n$, where $n$ is the number of points in the data set, since the more items there are, the more clusterings might theoretically exist.

Across each metric the function and general clusterings were similar, while the mechanism condition is a bit closer to the function than the general condition. These results offer preliminary support for our hypothesis. They not only suggest that participants are sensitive to mechanism, but also that their sorting conforms to three mechanism domains: mechanical, chemical, and electromagnetic.

## 3. Studies 2a and 2b: Abstract attribution

Schultz, Bonawitz, and Griffiths [36] investigated whether preschoolers would make causal attributions based on the domains of their naïve theories. Motivated by Griffiths and Tenenbaum's [37] theory-based causal induction theory, Schultz et al., pitted domain-specific evidence (e.g., theories) against domain-general statistical inference. In theory, children's judgments might emerge from evidence, theories, or a combination of the two. In any case, their judgments should depend on the strength of each component. According to Schultz et al. [36], domain-specific theory knowledge manifests in our prior probabilities of theories. However, they also show that children can pick up on statistical regularities that can override domain-specific theories.

**Table 1. Similarities between the mechanism, function, and general item clusterings (higher numbers mean greater similarity for the RI and ARI, while lower numbers mean greater similarity for the VI).**

|  | Mechanism | Function | General |
|---|---|---|---|
| Mechanism |  | RI = 0.77 | RI = 0.72 |
|  |  | ARI = 0.51 | ARI = 0.40 |
|  |  | VI = 0.64 | VI = 1.01 |
| Function |  |  | RI = 0.85 |
|  |  |  | ARI = 0.66 |
|  |  |  | VI = 0.61 |

**Table 2. Normed stimuli used in Studies 2a and 2b, alongside judged domain and chi-square tests for the three causes and the three effects in Studies 2a and 2b.**

| Intended Domain | Cause Item | Agreement with Intended Domain | Effect Item | Agreement with Intended Domain |
|---|---|---|---|---|
| Mechanical | This machine works by applying physical pressure. | 85.00% $X^2$ (2) = 24.10, p < 0.001 | Imagine a machine that modifies the shape of objects. | 90.00% $X^2$ (2) = 29.20, p < 0.001 |
| Chemical | This machine works by invoking a chemical reaction. | 95.00% $X^2$ (2) = 34.30, p < 0.001 | Imagine a machine that modifies the color of objects. | 85.00% $X^2$ (2) = 24.70, p < 0.001 |
| Electro magnetic | This machine works by emitting an electrical current. | 80.00% $X^2$ (2) = 19.60, p < 0.001 | Imagine a machine that modifies the temperature of objects. | 70.00% $X^2$ (2) = 12.40, p = 0.002 |

We presented participants with a series of effects and asked them to select the most likely cause. The domain-matching hypothesis predicts that people will select the cause that matches the domain of the effect. Study 2a tested adults and 2b tested children. As predicted by the domain-matching hypothesis, children were expected to select the cause that matched the domain of the effect. For each effect, participants were required to select only one cause. We wanted to know if children could select a cause that matched the domain of the effect, or whether their choices varied depending on their age across development.

### 3.1 Norming

Six items were crafted in total, three causes and three effects. Each item was designed with an intended mechanism domain in mind: mechanical, chemical, or electromagnetic. For example, the mechanical effect item was, "Imagine a machine that alters the shape of objects," and the chemical cause was, "This machine works by invoking a chemical reaction." The complete set of stimuli can be found in Table 2 below. The norming instructions given to subjects were as follows: "For each of the following statements, think about how the event of interest actually works. Think about the kinds of mechanisms involved, and then indicate whether you believe that the primary mechanisms are mainly mechanical, chemical, or energy-based (waves of energy/electricity) in nature." The last category is referred to as "electromagnetic" in subsequent studies.

On average, the proportion of participants whose judged domains matched our intended domains was high (84.2%). Chemical items were most agreed upon (90.0%) followed by their mechanical (87.5%) and electromagnetic counterparts (75.0%; see Table 2 below). For all items, the intended domain was always endorsed more often than either unintended domain. For each item, a chi-square test was performed to see if the item's judged domain was different from what would be predicted by chance given these three categories.

### 3.2 Method

In Study 2a, 51 adult participants (24 males, 27 females; Mage = 31.59 years, SDage = 9.85 years) recruited from Mechanical Turk completed the task. Participants took on average 2 minutes and 53 seconds to complete the task. In Study 2b, 88 children completed the online task using OptimalSort. The age range was between 8 to 17 (N = 88; 38 male, 46 female, 4 preferred not to say (Mage = 13.55 years, SDage = 3.07 years). On each trial, participants (in both Study 2a and 2b) were presented with one effect and were asked to rank the three candidate causes in descending order of likelihood. Participants did this for each effect. The questions and their choices were presented in random order.

**Fig 4. Proportion of responses for which the 1st–ranked cause matched the intended domain of each of the three effects (chance is $33^1/_3$%; error bars are 95% confidence intervals).**

## 3.3 Results & discussion

**3.3.1 Study 2a.** For each effect, the first-ranked causes were analyzed. In the mechanical effect condition, the mechanical cause was selected first 37 (73%) out of 51 times, $\chi^2$ (2) = 35.29, p < 0.001. In the chemical effect condition, 42 (88%) of 48 judgments (3 participants failed to make judgments) pitted the chemical cause as most likely, $\chi^2$ (1, N = 48) = 63.88, p < 0.001. And in the electromagnetic effect, the electromagnetic cause was selected first 26 (53%) of 49 times (2 participants failed to make judgments), $\chi^2$ (2, N = 49) = 8.61, p = 0.01. Overall, participants were more likely to select a cause that matched the domain of an effect (see Fig 4 and Table 3).

**3.3.2 Study 2b.** For the mechanical effect, 66 (75%) out of 88 participants appropriately selected the mechanical cause. For the chemical effect, 67 (76%) participants selected the

**Table 3. A frequency listing of all ranking data.**

| | Rank | Mechanical Effect | Chemical Effect | Electromagnetic Effect |
|---|---|---|---|---|
| Mechanical Cause | 1st | 37 | 5 | 11 |
| | 2nd | 10 | 7 | 5 |
| | 3rd | 4 | 36 | 33 |
| Chemical Cause | 1st | 7 | 42 | 12 |
| | 2nd | 27 | 3 | 25 |
| | 3rd | 17 | 3 | 12 |
| Electromagnetic Cause | 1st | 7 | 1 | 26 |
| | 2nd | 14 | 38 | 19 |
| | 3rd | 30 | 9 | 4 |

chemical cause. Similar patterns were observed for the electromagnetic effect, where 61 (69.3%) participants appropriately selected the electromagnetic cause. One-way ANOVAs confirmed nonsignificant age effects for each domain (mechanical, chemical, electromagnetic; p > .05, for all). The results indicated that children were more likely to select a cause that matched the domain of an effect. These results were similar to the adult data.

An alternative account of how people make these decisions is that they are sensitive to base rates: people select what they believe to be the cause with the highest base rate, regardless of the context that the given effect provides. Because we do not know the base rates of our causes, it is unclear what that would mean for these stimuli, so we tested the base rate account in Study 3.

Another account of how people come to attributions is that they base their responses on similarity [6]. For example, they might select the mechanical cause when it is most similar to the mechanical effect. However, the similarity relations in our stimuli do not clearly coincide with domain, at least if we understand "similarity" to refer to perceptual similarity. If we allow "similarity" to include non-perceptual elements, then it becomes too unconstrained to be helpful. For example, perhaps the application of pressure is more similar to shape modification than color change. However, is emitting an electrical current more similar to temperature or color change? In most cases, electrical currents are invisible; and to a lesser extent the same is true for temperature fluctuations. If people are basing their judgement on the fact that temperature changes tend to associate with electromagnetic activity, then they are relying on the similarity of domains, not on the perceptual similarity of events.

Another account might predict that people make these judgments only after a theoretical evaluation of the mechanisms involved across each pair of cause and effect. According to this "knowledge hypothesis," people would respond in kind not because of mechanism domain matching, but because they have a complete-enough understanding of actual mechanisms to derive a response based on a mental simulation or similar computation. The fact that the children's data largely replicated the results of the adult data argues against this account. Presumably the ability to generate a mental simulation of a complex process develops over time. It is less likely for children to have a complete-enough understanding of artifact mechanisms. We conclude that the best explanation for the data is the proposed domain-matching heuristic.

## 4. Study 3: Abstract prediction

Study 3 was designed to mirror Study 2 but in the causal direction, the more natural direction to reason in [38–40]. The domain-matching hypothesis is agnostic with respect to causal directionality, and so the prediction here is the converse: that people should prefer effects that come from the same domain as their cause.

### 4.1 Method

51 participants (32 males, 19 females; M$age$ = 31.31 years, SD$age$ = 9.03 years) recruited from Mechanical Turk completed the task over an average of 3 minutes and 20 seconds. The materials from the previous two studies were reused. This time, participants were first presented with a cause, and then asked to rank the three effects in descending order of likelihood. Participants repeated this for each effect. The three questions and their choices were presented in random order.

### 4.2. Results & discussion

For the mechanical cause, the mechanical effect was selected first 34 (67%) times out of a total of 51 judgments, $X^2(2) = 25.76$, $p < 0.001$. In the chemical cause condition, 35 (69%) of 51

judgments indicated the chemical effect was most likely, $X^2(2) = 31.53$, $p < 0.001$. For the electromagnetic cause the electromagnetic effect was selected first 36 (71%) of 51 times, $X^2(2) = 31.88$, p < 0.001. We again take this as evidence supporting the domain-matching heuristic. This experiment rules out the idea that people are choosing the option with the highest base rate. That hypothesis predicts that a single effect would be chosen more often than the others for all items.

## 5. Studies 4a and 4b: Concrete attribution

Studies 2 and 3 asked participants to reason about relatively abstract items. Study 4 also asked participants to perform causal attribution, but with a larger number of more diverse items. This study serves to compare the predictions of our mechanistic domain(s) hypothesis to those of statistical and knowledge-based accounts directly.

Study 4 test items were designed so that causes and effects that matched mechanism domains would be objectively or subjectively counter-normative (i.e., in contradiction with statistical or theoretical knowledge or both).

We normed a set of triplets composed of an effect, a within-domain cause, and a cross-domain cause. Like Study 2, participants were presented with an effect and asked to choose the likelier cause. During our norming phase, we also collected likelihood judgments for all events. A sample triplet including mechanism domain and likelihood obtained during norming is presented in Table A1 in S1 Appendix.

We predicted that people would select causes that match the mechanism domain of the effects (i.e., within-domain) causes more often than ones that do not match the domain of the effects (i.e., cross-domain) causes. Because cross-domain causes were chosen that were more likely than within-domain causes, our predictions diverge from those based on the prevalence of causes or covariation of causes and effects (which would suggest, for example, that cars are likelier to be in accidents than they are to have their batteries short-circuit). It also implies that people will often respond contrary to what a knowledge-based account would dictate. For example, in another test item we ask participants to determine whether a house fire is more likely to be the result of leaving a wool sweater by the lit fireplace or plugging an air conditioning unit into an extension cord. Though people might have assumed that leaving the sweater is the more likely culprit, extension cord fires are a common cause of houses burning down, while wool is actually used in airplane upholstery because of its fire-retardant properties. Our predictions imply there is more to consider in causal attribution than base rates and mechanistic knowledge. Study 4a examined adult judgments and 4b children's judgments.

### 5.1 Method

**5.1.1 Norming.** Norming was conducted over two sessions. In the first session, a group of participants was only asked about causes. In the second, another group of participants was asked about effects. For session 1, 60 participants (28 males, 32 females; Mage = 34.12 years, SDage = 12.77 years) recruited from Amazon's Mechanical Turk via TurkGate [41] completed the task. Participants were paid twenty cents for participation, which on average 6 minutes and 1 second. Session 1 asked half of the participants (N = 30) to make likelihood judgments for statements like, "The rotor inside a random fan broke," and, "A person spilled bleach on their sweater". The other half of participants (N = 30) made domain judgments for the same items (i.e., mechanical, chemical, or electromagnetic). The full instructions were, "For each of the following statements, think about how the event of interest actually works. Think about the kinds of mechanisms involved, and then indicate whether you believe that the primary

mechanisms are mainly mechanical, chemical, or energy-based (waves of energy/electricity) in nature."

Session 2 was conducted between two other, unrelated studies on political psychology. 96 participants (35 males, 59 females; Mage = 35.57 years, SDage = 12.27 years) recruited from Amazon's Mechanical Turk via TurkGate [41] completed the task. Participants were paid sixty cents for participation in a larger group of surveys. In session 2, participants received the same instructions and questions as in session 1. However, they were asked about the effect items rather than the cause items. One group of items was eliminated due to a typo. The final set of 57 test items alongside their likelihood and domain judgments can be seen in Table A1 in S1 Appendix.

Table A1 in S1 Appendix shows the mean and standard deviation of likelihoods for each item. In addition, the proportion of participants who selected each mechanism domain is given, along with a Chi-Square test (df = 2) to determine whether responses were significantly different from chance. The last column represents the same inclusion criterion used in the previous experiment, which indicates whether more than 1/3 of participants agreed with the items' intended domain.

### 5.2 Study 4a

30 adult participants (19 males, 11 females; Mage = 35.47 years, SDage = 11.81 years) recruited from Mechanical Turk completed the task. Participants took on average 5 minutes and 18 seconds to complete the study. Participants were presented with effect items from the mechanical, chemical, and electromagnetic mechanism domains, each of which was paired with a cause that was within-domain (e.g., mechanical causes mechanical) and a cause that was cross-domain (e.g., chemical or electromagnetic causes mechanical). For each effect presented, participants were asked to choose which of the two causes they believed was responsible for the effect. All items were presented in random order. (see Table A1 in S1 Appendix) The full instructions were, "Thank you for participating! Please respond to the following statements according to your beliefs. For each statement choose the reason that you feel is more likely. Read all of the text but do not hesitate too long on any particular response."

### 5.3 Study 4b

In total 88 children completed the task online. Participants responded to 9 questions presented with triplets following the same testing protocol as adults. The triplet items contained an effect, a within-domain cause, and a cross-domain cause as shown in Table 4. As in Study 4a, children were expected to follow the domain-matching hypothesis and select within-domain causes more often than cross-domain causes.

### 5.4 Results & discussion

**5.4.1 Study 4a.** As predicted, participants selected the within-domain cause more often than the cross-domain cause, $\chi2 (1) = 20.15$, $p < 0.001$ (see Fig 5).

We also predicted that participants would choose the within-domain causes even when those causes did not obtain the highest likelihood judgments obtained during norming, which were on average higher for cross-domain items (Mcross = 64.90%, SDcross = 33.61%) than within-domain ones (Mwithin = 57.76%, SDwithin = 34.42%; $t(838) = -3.04$, $p = 0.002$). Participant's attributions were typically in disagreement with any theory-based response as well.

**5.4.2 Study 4b.** Children selected the within-domain cause more often than the cross-domain cause. The most challenging triplets were the ones which required participants to make a choice between the mechanical and chemical cause (see Table 5). Further one-way

**Table 4. Triplets.**

| Items | Intended Domain | % |
|---|---|---|
| 1. Tim couldn't put the square block in the round hole | M | |
| The square was blue, and the round hole was green | C | 3.4 |
| The square was bigger than the round hole. | M | 96.6 |
| 2. John's car wasn't very shiny after treating it with wax. | C | |
| John used the wrong kind of wax when waxing his car | C | 64.8 |
| John used the wrong waxing motion when waxing his car. | M | 35.2 |
| 3. Alfie made a bread, but he found it to be smaller than he hoped. | C | |
| Alfie didn't mix the dough enough when making bread. | M | 54.5 |
| Alfie used too little yeast when making bread. | C | 45.5 |
| 4. The radio in Hannah's car no longer works. | E | |
| Hannah's car was in an accident. | M | 4.5 |
| Hannah's car's battery short-circuited. | E | 95.5 |
| 5. Jem's cell phone battery doesn't hold charge as well as it used to. | E | |
| Jem's phone's vibrated during incoming calls. | M | 9.1 |
| Jem's phone's battery never fully drained. | E | 90.9 |
| 6. Tom's house caught on fire. | C | |
| A fireplace was lit with a wool sweater nearby. | C | 67.0 |
| A random person plugged an air conditioner into an extension cord. | E | 33.0 |
| 7. Jack got sick after using a dirty restroom. | C | |
| Jack rubbed his hands together under running water when cleaning his hands. | M | 50 |
| Jack dipped and left his hands in soapy water before rinsing them off when cleaning his hands. | C | 50 |
| 8. The fan inside Tim's laptop suddenly slowed down. | M | |
| Tim's laptop was kept in a very dusty room. | M | 70.5 |
| Tim's internet cut off while watching a movie. | E | 29.5 |
| 9. Sara's sweater got a hole in it. | M | |
| Sara spilled bleach on her sweater. | C | 6.8 |
| Sara's sweater got caught on her belt's fastener. | M | 93.2 |

ANOVAs for each triplet indicated nonsignificant age effects on participants' choices after the Bonferroni correction (p>.05 for all except for the sixth triplet, which was p = .033).

To examine whether there was a significant difference between the percentage of children who chose within-domain causes and the percentage who chose cross-domain causes, we used a chi-square test (testing the proportions against the null hypothesis). As can be seen in Table 5, the 3rd and 7th triplet values were nonsignificant. All other triplets showed a significant difference in participants' choices favoring within-domain causes.

For the first triplet, the regression model significantly explained 40.8% (Nagelkerke R2) of the variance in children's responses and correctly classified 96.6% of cases, where age effect was non-significant. For the second triplet the model explained 5% of the variance and correctly classified 67% of cases, where age effect was non-significant. Similarly, for the rest of the triplets the analyses indicated non-significant age effects in accord with the ANOVA results (see also Fig 6, showing participants' choices across the age groups on the triplets used in Studies 4a and 4b).

Both adults and children selected the within-domain cause more often than the cross-domain cause, suggesting their choices were based on the domains of the mechanism. The results could be due to participants considering causes from the same domain to have a higher

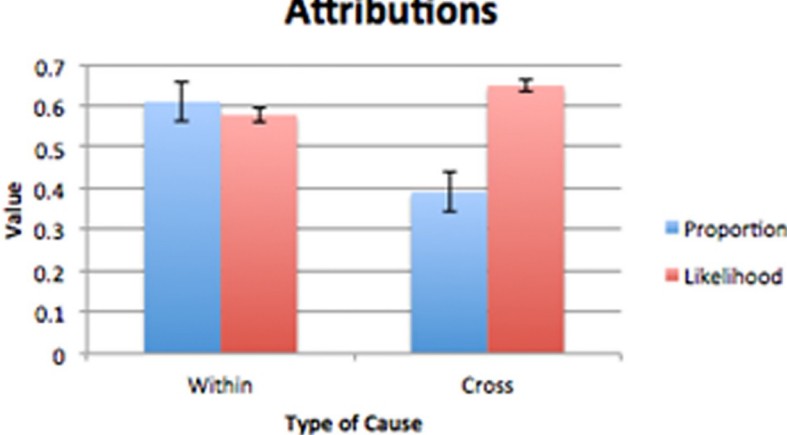

**Fig 5. Proportion of attributions (within and cross proportions sum to 1) and mean likelihoods for causes that matched the domain of the effect (within-domain) vs. those that did not (cross-domain).** (Error bars are 95% Confidence Intervals for the proportions).

causal strength in generating their corresponding effects. To shed light on this issue, we ran Study 5, presenting participants with multiple effect items that did not always favor causal strength.

## 6. Study 5: Attribution with stable causes

Is it possible that something altogether different from base rates, causal knowledge, similarity, or mechanism domains is driving behavior? Maybe people are responding to another cue. For example, consider the test item: "A person's sweater got a hole in it." The corresponding, cross-domain cause is, "The person spilled bleach on their sweater," while the within-domain cause is, "The person's sweater got caught on their belt's fastener." For the sake of completeness, we test the possibility that some people are endorsing the within-domain cause not because it matches the mechanical mechanism domain, but because they believe belt fasteners are highly causative for some reason. One way to test such factors would be to use the same causes across different effects (i.e., to manipulate effects while holding causes stable). Would the same people endorse belt fasteners regardless of the effects they are presented with? If in

**Table 5. Chi-square and logistic regression tests for each triplet.**

| Triplet | N | χ2 tests for each triplet | | | | Logistic regressions | | |
|---|---|---|---|---|---|---|---|---|
| | | Mean | SD | χ2 | Sig | Nagelkerke R$^2$ | Wald test for age | Sig |
| 1 | 88 | 1.97 | .183 | 76.409 | .000 | .408 | 2.195 | .138 |
| 2 | 88 | 1.35 | .480 | 7.682 | .006 | .050 | .393 | .531 |
| 3 | 88 | 1.45 | .501 | .727 | .394 | .019 | .591 | .442 |
| 4 | 88 | 1.95 | .209 | 72.727 | .000 | .252 | .174 | .677 |
| 5 | 88 | 1.91 | .289 | 58.909 | .000 | .076 | 1.712 | .191 |
| 6 | 88 | 1.33 | .473 | 10.227 | .001 | .067 | .071 | .789 |
| 7 | 88 | 1.50 | .503 | .001 | 1 | .062 | 1.841 | .175 |
| 8 | 88 | 1.30 | .459 | 14.727 | .000 | .018 | .166 | .683 |
| 9 | 88 | 1.93 | .254 | 65.636 | .000 | .292 | .025 | .875 |

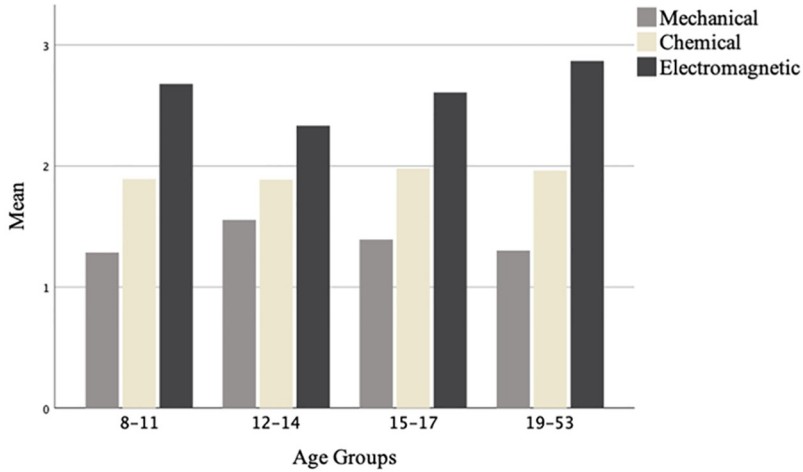

**Fig 6. Participants' choices across age groups.**

these cases people still select the within-domain cause more often, additional support will be provided for the domain-matching hypothesis.

To test this, we developed a new set of quadruplets (rather than triplets) as shown in S2 Appendix. For example, one quadruplet's effects were, "A person's sweater got discolored," and, "A person's sweater got holes in it." The quadruplet's causes were, "A person poured bleach over a stain on their sweater," and, "A person rubbed steel wool over a stain on their sweater." Bleach can discolor a sweater, but it can also create holes by dissolving the sweater's fabric. Given the effect of "holes in a sweater," the domain-matching prediction is that participants would more often select the steel wool cause, even though, again, it might be less probable (steel wool likely does more damage than good in removing common stains). At the same time, the steel wool cause is predicted to elicit a different response given the discoloration effect. This study relied on additional norming to establish a set of items whereby each cause had a validated within- as well as a cross-domain effect, and vice versa. Study 5 asked participants to make a series of causal attribution once again. The prediction again was that participants would select the within-domain cause more often than the cross-domain cause.

## 6.1. Method

**6.1.1. Norming.** This study was conducted between two other, unrelated studies on political psychology. 100 participants (41 males, 59 females; Mage = 36.49 years, SDage = 12.75 years) recruited from Amazon's Mechanical Turk via TurkGate [41] completed the task. Participants were paid sixty cents for their participation in a larger group of surveys. Participants were entered into one of two conditions, such that any given participant either made a series of likelihood judgments or domain judgments, where all items were presented in random order. Judgment results are provided in S2 Appendix. According to the same 1/3 criterion used in previous studies, judged domain matched intended domain for nearly all (92.86%) items. The two Cause within-Cause cross-Effect within-Effect cross quadruplets that did not completely meet the 1/3 criterion are in italics in the Table A2 in S2 Appendix.

Furthermore, one hundred participants (44 males, 56 females; Mage = 32.95 years, SDage = 11.17 years) recruited from Mechanical Turk completed the task. Participants completed the task within a larger group of surveys. In this study, participants were provided with

an effect and asked to determine the likelier cause. Any given participant was presented with only one of the two effect items per quadruplet.

## 6.2. Results & discussion

Although we used the same causes across different effects, participants selected the within-domain cause significantly more often (63.00%) than the cross-domain cause (37.00%), $\chi^2$ (1) = 33.80, p < 0.001. Thus, the same causes and effects elicited different responses depending on their context (e.g., corresponding effects and causes). This finding also discredits both the base rate hypothesis, and the idea that idiosyncratic properties of select events engender stable response patterns.

We take this as some evidence that the causal strength of each cause for its corresponding effect is higher when the causes match the domain of the effect; we see greater causal strength when domains match. While it could be that the results occurred because causes that matched the domain of their effect just happened to have greater causal strength, this would be an unlikely co-incidence. While it is possible that each of our items suffers from a different confound, the one invariant across items is that causal strength is higher when domains match, suggesting that commonality of domain is the operative variable.

## 7. General discussion

This study investigated whether people tend to think about the mechanisms that relate causes to effects as sitting within one of small number of domains, whether people use a domain-matching heuristic. In total 5 studies with adults and 2 with children looked for evidence of the phenomenon. The first study required participants to cluster artifacts, and we identified specific domains that people commonly employ in their clusters. Using these domains, the next studies evaluated the domain-matching heuristic by testing predictions about causal attribution, prediction, judgement, or subjective understanding.

The results suggested that people do use knowledge of mechanism domains when engaged in various causal reasoning tasks. People's judgments in attribution, prediction, and believability abide by the domain-matching hypothesis, which states that causal perception is enhanced when cause and effect share a common mechanism domain. These findings cannot easily be explained by an appeal to base rates, mechanism knowledge, or similarity between events alone.

The mechanism domains we found evidence for were the mechanical, chemical, and electromagnetic domains. Even though preliminary evidence supported these domains in particular, the domain-matching hypothesis is not committed to any specific domains, and there very well may be others, such as those in the social realm. These were excluded in order to restrict scope. Other researchers have distinguished biological causal explanations from explanations for artifacts and social entities [29, 42]. Others focus on how teleological explanations differ from causal ones [16, 43–45]. Our studies are limited to artifacts. Within that broad domain, our studies suggest a basic distinction among mechanical, chemical, and electromagnetic forms.

Mechanism domains seem to help people make sense of an otherwise complex world. They possibly allow us to bypass deep understanding in order to come to satisfactory conclusions in a cognitively economic manner. We propose domain matching is a heuristic we employ in order to reduce search spaces during causal reasoning, and though it can pick up on the veridical structure of the world, it can also lead us astray (albeit in predictable directions).

Developmental data supported this interpretation and revealed that, despite limited experience with the artifacts, children were more likely to attribute a causal relation when two events

shared a mechanism domain. Their choices were substantially above chance, similar to that of adults. This implies that, even when lacking mechanistic knowledge, the ability to reason about causal relations and linking mechanisms supporting these relations has already matured into adult form sometime around the age of 11. We leave open questions about the manner and extent to which much younger children use the domain-matching heuristic. The data we report here make only a small incursion into our understanding of how people develop their extraordinary ability to understand the causal powers of everyday objects.

## Supporting information

**S1 Appendix. List of items used in Study 1, and Table A1.** Norming data for study 4a and 4b.
(DOCX)

**S2 Appendix. Table A2.** Norming data for Study 5.
(DOCX)

**S1 Data.**
(ZIP)

## Author Contributions

**Conceptualization:** Selma Dündar-Coecke, Gideon Goldin, Steven A. Sloman.

**Data curation:** Selma Dündar-Coecke, Gideon Goldin.

**Formal analysis:** Selma Dündar-Coecke, Gideon Goldin.

**Funding acquisition:** Selma Dündar-Coecke, Steven A. Sloman.

**Investigation:** Selma Dündar-Coecke, Gideon Goldin, Steven A. Sloman.

**Methodology:** Selma Dündar-Coecke, Gideon Goldin, Steven A. Sloman.

**Project administration:** Gideon Goldin, Steven A. Sloman.

**Resources:** Gideon Goldin, Steven A. Sloman.

**Software:** Gideon Goldin.

**Supervision:** Steven A. Sloman.

**Validation:** Gideon Goldin.

**Visualization:** Gideon Goldin.

**Writing – original draft:** Selma Dündar-Coecke, Gideon Goldin, Steven A. Sloman.

**Writing – review & editing:** Selma Dündar-Coecke, Gideon Goldin, Steven A. Sloman.

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
