## [Decision Letter · Decision Letter 0]

13 Jul 2021

PONE-D-21-08341

Causal reasoning without mechanism: The domain-matching heuristic

PLOS ONE

Dear Dr. Dündar-Coecke,

thank you for submitting your manuscript to PLOS ONE.  After careful consideration by three experts in the field, we feel that it has merit but does not fully meet PLOS ONE’s publication criteria as it currently stands.  I appreciate the Reviewers' efforts while evaluating the manuscript and their thoughtful and detailed comments on the work.  As you will see, the Reviewers have found much interest in your work.  We invite you to submit a revised version of the manuscript that addresses the points raised during the review process, paying special attention to the following issues:

1) Please thoroughly address all technical and statistical issues raised by the Reviewers below.

2) Please report if the data sets were tested for normality and which statistical inference was used for respective data including test directionality and multiplicity correction where applicable. 

3) Two Reviewers have mentioned some issues with data availability.  Please clarify; in case of uncertainty, please contact PLOS ONE Editorial Team on this occasion.

4) Although an Ethics board approval was obtained, please indicate in the main text if/that the written consent was given by the participants.

Please also make sure to address all the other Reviewers' comments in the point-by-point manner.

Please submit your revised manuscript within six months from this date as thereafter, any revision has to be considered a new submission.  If you will need more time than this to complete your revisions, please reply to this message or contact the journal office at plosone@plos.org. Please include the following items when submitting your revised manuscript:

We look forward to receiving your revised manuscript.

Thank you for choosing PLOS ONE for communicating your research.

Kind regards,

Sasha

Alexander N. 'Sasha' Sokolov, Ph.D.

Academic Editor

PLOS ONE

Journal Requirements:

2. You indicated that you had ethical approval for your study. In your Methods section, please ensure you have also stated whether your IRB specifically approved the method of parental consent.

4. We note you have included a table to which you do not refer in the text of your manuscript. Please ensure that you refer to Table 2 in your text; if accepted, production will need this reference to link the reader to the Table.

Reviewers' comments:

Reviewer's Responses to Questions

**Comments to the Author**

1. Is the manuscript technically sound, and do the data support the conclusions?

Reviewer #1: Yes

Reviewer #2: Partly

Reviewer #3: Partly

2. Has the statistical analysis been performed appropriately and rigorously? 

Reviewer #1: Yes

Reviewer #2: Yes

Reviewer #3: Yes

3. Have the authors made all data underlying the findings in their manuscript fully available?

Reviewer #1: No

Reviewer #2: Yes

Reviewer #3: No

4. Is the manuscript presented in an intelligible fashion and written in standard English?

Reviewer #1: Yes

Reviewer #2: Yes

Reviewer #3: Yes

5. Review Comments to the Author

Reviewer #1: This paper examines a longstanding puzzle in cognitive psychology: On the one hand, many results point to the importance of mechanism knowledge in causal reasoning; yet on the other hand, people are remarkably bad at articulating these mechanisms, suggesting that mechanism knowledge is shallow and skeletal. This paper presents evidence for a domain-matching heuristic, such that people categorize causal mechanisms into broad domains (e.g., mechanical, chemical, electromagnetic), matching causes and effects by domain despite a lack of detailed knowledge of how these mechanisms work. Several experiments in both adults and children support this hypothesis.

Overall, I enjoyed reading this paper and think it makes a contribution to the literature on causal cognition. Here are a few questions I had while reading it, mainly aimed at further understanding the authors’ theory and how it fits in with other perspectives.

1. The coverage in the introduction of the literature on mechanism knowledge in causal reasoning is generally very good. For readers who are unfamiliar with phenomena such has the illusion of explanatory depth and the related results from the authors’ own research group, it would be useful to devote a couple more paragraphs to summarizing the evidence that people have impoverished knowledge of causal mechanisms.

2. One relevant literature the authors do not consider is related work on other similarity-matching heuristics in causal reasoning. Off-hand, I can think of at least four papers in this vein that the authors might find useful in terms of explaining and situating their theory:

Einhorn, H. J., & Hogarth, R. M. (1986). Judging probable cause. Psychological Bulletin, 99(1), 3-19.

Johnson, S. G. B., & Keil, F. C. (2014). Causal inference and the hierarchical structure of experience. Journal of Experimental Psychology: General, 143(6), 2223.

LeBoeuf, R. A., & Norton, M. I. (2012). Consequence-cause matching: Looking to the consequences of events to infer their causes. Journal of Consumer Research, 39(1), 128-141.

Lim, J. B., & Oppenheimer, D. M. (2020). Explanatory preferences for complexity matching. PloS one, 15(4), e0230929.

Work on the “laws of sympathetic magic” is also broadly relevant, although less directly about casual reasoning in particular:

Rozin, P., Millman, L., & Nemeroff, C. (1986). Operation of the laws of sympathetic magic in disgust and other domains. Journal of Personality and Social Psychology, 50(4), 703–712.

It would be useful if the authors could explain how their similarity-matching principle relates to the ideas in these papers. Does it share a similar adaptive or informational logic?

3. I am a little confused about the authors’ use of terminology surrounding “domains.” This is not entirely on the authors, as this terminology is somewhat slippery in the broader literature as well.

(i) *Categories* In a somewhat loose sense, sometimes people use the word “domain” to simply refer to different categories of knowledge or phenomena. For example, professors at a music conservatory might have different “domains” of knowledge such as music history, music theory, and music performance. In that sense, the categories studied in this paper (e.g., electromagnetism vs. mechanics) surely count as domains. It seems like there’s nothing inherently special about these particular categories, but rather these are the level of categories that participants naturally learned. In the vein of work by Coley, Medin, and others on expertise and categorization, these categories could easily shift with increased knowledge and likely would become more specific with more experience.

(ii) *Modules* In its more technical sense, “domain” refers to something more like a mental module. For example, folk-psychology or folk-physics could be a domain because core knowledge about these domains plausibly evolved, whereas folk-music-history would not be a good candidate for a domain, nor (I think) would folk-electromagnetism. Presumably modules (to the extent they exist) have not changed over the course of human history, but I am sure that everyday intuitions about electromagnetism have changed very much over the past 100 years.

As you can see, I think the authors are basically using the word “domain” synonymously with “category,” but this is causing me some confusion as there are some references to the domain-specificity literature that make it seem like the authors are sometimes using some of the theoretical apparatus of modules.

Related – and perhaps this would help with the above point – the discussion of Sloman, Lombrozo, and Malt’s analysis is not integrated as well as it could be with the other points made in the introduction. I like the Sloman et al. chapter a lot and find it a good way of summarizing various writers’ views on this topic, and I also find those authors’ “mild ontology” view plausible. But the relationship between mild ontology and the “domain matching” hypothesis isn’t particularly clear, mainly because “domain specificity” or “extra strong ontology” is very much along the lines of modules (option ii), whereas “mild ontology” is much more along the lines of categories (option i).

Basically, I would like the authors to better articulate what they mean by “mechanism domains,” and possibly consider using a clearer term that does not carry as much baggage and ambiguity.

4. One thing that made the paper a little bit hard to read was the introduction of alternative explanations throughout the presentation of the results rather than in the front-end of the paper. Could the authors add a section that collects the alternative accounts in one place (e.g., similarity, base rates, mechanism knowledge) and explains the kind of evidence that will be important for addressing them? These alternatives are not explained in very much detail in the current version of the manuscript which makes it somewhat difficult to evaluate their plausibility. In particular, the similarity account seems like a plausible alternative that should be discussed more thoroughly, both in the introduction and discussion, as fleshing this out would likely help to clarify the authors’ own proposal.

5. Overall I found the package of studies to be convincing. I had just a couple suggestions around the presentation:

(i) The results of Study 1 sound very interesting but they are only described qualitatively. Could the authors provide some visualization of how participants sorted the items?

(ii) Could the authors present figures that show the age effects (including comparisons to adults when the data are comparable)? Even though age effects were not substantial, the developmental aspect of these studies will be interesting to many readers.

Minor:

- The numbering of the tables is confusing, as the tables are numbered sequentially between the main text and appendix. Referring to the appendix tables as A1 and A2 (or simply as Appendix 1 and 2) would be clearer.

- Some of the references to tables seem to refer to the incorrect table, including lines 446, 474, and 521… generally throughout Studies 4 and 5 the relationship between the main text and tables is confusing.

Thanks very much for the opportunity to read your work, and best of luck with next steps!

Sam Johnson

(I sign all reviews)

Reviewer #2: Review of “Causal reasoning without mechanism” by Dündar-Coecke, Goldin, and Sloman

In this paper, Dündar-Coecke, Goldin, and Sloman (DGS henceforth) investigate how reasoners (human adults and children) assess the causal status of co-occurred events in the absence of specific mechanistic or covariational knowledge about the corresponding types of events. The main hypothesis DGS propose in this paper is that people assess causal links between events in such cases by considering abstract mechanism domains. More specifically, the authors propose and test the idea that reasoners are more likely to regard certain events as causally connected if these events belong to the same global mechanism class rather than to different mechanism classes. The authors first established (Experiment 1) and then tested (subsequent studies) three global mechanism categories: (1) mechanical, (2) chemical, and (3) electromagnetic. For both adults and children, it was found that causal judgments, irrespective of whether they were made in the predictive or the diagnostic direction (Experiments 2 and 3), corresponded to the proposed domain matching hypothesis: when selecting the most probable cause (or effect) of a target event, subjects tended to select an event that belonged to the same mechanism category. Experiment 4 replicated the effect with test times (cause and effect events) that were more concrete than the rather abstract items used in Experiments 2 and 3.

I think that this is an interesting topic and that DGS propose a, in my view, very plausible hypothesis. I find it quite natural, for example, that if a reasoner observes an event and tries to uncover its cause(s), he or she will search for events that could plausibly be causally relevant. I share the authors’ intuition that broad mechanism categories are a useful guide in this respect. I also think that the question how people form causal beliefs in the absence of directly relevant information, such as information about covariation, has been understudied. However, although I’m quite enthusiastic about DGSs’ research, I think that some more work needs to be done before this manuscript can turn into an impactful publication. Some of my remarks, that can probably be dealt with fast, concern the flow and the presentation of ideas and conclusions, and the lacking of some information in the empirical sections, but I also think that an additional final experiment could be necessary to have a more convincing package.

Flow of ideas and structuring:

My impression is that the main theory of the paper is told relatively straightforwardly, but I also was a bit confused by some of the paragraphs in the introduction and I think that many of the paragraphs are not very well connected. I fear that this tends to impair the reading flow. For example, in section 1.1 “Causal attribution in the absence of mechanism knowledge” it was not really clear to me what the authors wanted to tell the reader. If the purpose here was to convince the reader that reasoners tend to be lacking specific mechanism knowledge in many domains but often seem to be guided by a more abstract sense of mechanisms, that this could be done more concisely. I really wondered, for example, why the paragraph on page 5 beginning in line 108 was in there. The paragraph starts with “`The domain-based difference’ hypothesis, proposed by [17], suggests that domains can indeed play functional role even after equating strength of belief. In that study, a sample of people were asked several questions from the scientific and religious domains […]” and ends with “[…] This showed that people think a mechanistic explanation is a stronger requirement in science than religion.” I don’t understand why this is relevant for the present paper? I had the same question for the paragraph bout Rozenbit and Keil’s results starting in line 145, in which DGS summarize that people’s confidence in their understanding of devices decreases, for example, after they had tried to offer an explanations of how these devices work. Again, I think the point that DGS want to make in their theory section is that people do not seem to rely on specific mechanism knowledge, but this doesn’t mean that they are not guided by more abstract mechanism knowledge. I think the different paragraphs need to be connected better. Right now, it feels a bit like loose pieces and the main message emerges to the reader only somewhat implicitly.

Presentation of empirical results:

I think it would be appropriate to give some rationale for the sample sizes that the authors employed in their studies. Did the authors do an a priori planning of their studies? Are the sample sizes based on pilot or previous studies?

In the results sections when the authors present the means, I think it would be good to also report confidence intervals for the means and not just standard deviations. A related point is that, in Figure 2, the authors present 95% Cis for the proportions but SEs for the Likelihoods. Why? I think this should be the same for the bars in that figure. Since most readers will probably not expect different types of error bars within the same graph, a spontaneous impression (until one reads the legend) is that the proportion judgments were estimated much less precisely than the likelihood judgments, but this was actually not the case. Also, in Figure 1, the reader is not told what the error bars are. I think it would be best to have 95% CIs all the time.

Paragraph beginning in line 253:

DGS wrote “[…] their sorting conforms to the predicted mechanism domains: mechanical, chemical, and electromagnetic.” I think these domains were not predicted in the introduction. I was wondering how the authors can claim here that they found the predicted types of categories given that the goal of that study and the conducted cluster analysis was to identify relevant domains?

Items shown in Table 4:

I was wondering whether at least some of the items would have led to totally different results if they had been formulated slightly differently. Take, for example, Item 3: “Alfie made a bread, but he found it to be smaller than he hoped.” The different cause items are “Alfie didn’t mix the dough enough when making bread.” and “Alfie used too much yeast when making bread”. Subjects found the first option to be more likely, but what would have happened if the second option had mentioned “too little” rather than “too much” yeast. Using too little yeast leads to smaller breads, so wouldn’t this have changed subjects selections?

The items shown in Table 4 bring me to my biggest concern, which I think might have to be addressed in a final separate study. While I think that the authors convincingly address the potential base rate problem, another problem that I see is that subjects might have thought about items and might have based their selections on considerations of causal strength. Take for example, Item 9 about hole in the sweater, which is supposed to be an event belonging to the mechanical domain. The two potential causes are that bleach was spilled on the sweater and (chemical cause) and the sweater got caught on her belt’s fastener (mechanical). Most subjects apparently chose the “belt fastener”. Apart from belonging to the same domain, doesn’t this cause also have a higher causal strength in leading to this effect? I think it is statistically more likely that a sweater carries away a hole if I it gets tangled up in the belt faster than when it gets treated with bleach. I think most people who use bleach will not use so much that holes results, while an accidental rip of a pullover on the belt fastener is not unlikely to lead to a hole. An even more drastic example is Item 1 about the square that doesn’t fit through a round hole. The potential causes are that the square was to large for the hole and that their colors didn’t match. Most people know that color is causally irrelevant, while size is not. This knowledge, I think, is very concrete and I don’t think subjects’ decisions here were guided only by means of abstract domain matching.

I think that DGSs’ results would be more convincing if they controlled for the causal strength between the potential causes and the effect. While controlling for base rate, I think that it might be a possibility that subjects might have based their decisions on causal strength considerations. I see that DGS aimed to address this concern in their Study 5 (intended to control for how “causative” one event is for the other), but I don’t think that the way they did this is compelling. The idea in Study 5 was to control for how “causative” causes are by pairing them with different effects. For example, the cause item “a person rubbed steel wool over their sweater” was once coupled with the effect item “A person’s sweater got discolored” and once with the effect item “A person’s sweater got holes in it”. Another alternative cause item that was also used was “A person poured bleach over a stain on their sweater”. In the main task, subjects were presented with one of the effect items and asked to select the most likely cause. It was found, for example, that subjects who saw the effect item “A person’s sweater got holes in it” tended to select the mechanical cause item “A person rubbed steel wool over their sweater”, while they tended to select the cause item “A person poured bleach over a strain on their sweater” when the effect item was “A person’s sweater got discolored”. My concern is that subjects here simply selected the causes that have a higher causal strength in generating the presented target effects. Spilling bleach over a sweater will rarely lead to a hole in a sweater, whereas the causal strength between “rubbing a sweater with steel wool” and “making a hole in the fabric” is probably much higher. The reverse is true for the “discoloration” effect. DGS write that if one of the causes is more “causative” then subjects’ preference shouldn’t change depending on the effect item. But causal strength is relative. One and the same cause can have high causal strength with respect to one type of effect and weak (or zero) causal strength with respect to another. I was also wondering whether a causal strength theory can actually be dissociated from the domain matching theory that DGS propose. If not, I think what would be needed if causal strength and domain matching always make similar predictions is an experiment that shows that subjects do not think of causal strength (although it would lead to the same behavior if they did) when searching for the likeliest cause, but really about broader mechanism classes.

Minor comments:

- Line 153: I think a paragraph shouldn’t start with “So…”

- Line 153: “[…] gab between, on one hand, […]” should be “on the one hand”

- Line 175: “This implies that mechanism domains are not parts of a […]” should be “part” not “parts”

- Line 185: “Study 2 and 3 tested one […]” should be “Studies 2 and 3 tested one […]”

- Line 187: “[…], study 4 and 5 […]” should be “Studies 4 and 5 […]”

- Line 211: “[…] size and already constucted […]” should be “constructed”

- Line 313: “In the mechanical effect, the mechanical cause […]” should be sth. like “In the mechanical effect condition, […]”

- Line 341: “[…], so we test the base rate account in Study 4.” Should be “tested”

- Line 488: “As predicted, people selected […]” should be “subjects”

- Line 497: “[…] causes even if when […]” should be either “if” or “when”

In the reference list, many journal names were abbreviated “Cog Psyc.” or “Phil of Sci,” and I wondered whether this format is correct.

All in all, I think that this is an interesting project and I think that this work can make an interesting contribution. However, I think that the writing requires some polishing; my main point here is that I think that the paragraphs are often to loosely connected. I also think that DGS need to address the causal strength argument, although I unfortunately can’t suggest a specific experiment right now that would address this issue straightforwardly.

Signed,

Simon Stephan

Reviewer #3: This paper examines whether adults and children use a "mechanism domain" heuristic when connecting causes to effects. They find that people tend to associate independently-validated 'mechanical', 'chemical', and 'electromagentic' effects with causes in the same domain, and vice versa.

I have several concerns about Study 1, but I also believe that it could be either substantially expanded or removed from the paper and the end result would be suitable for publication in PLoS ONE.

First, the main issue with Study 1 is simply the lack of direct data available to readers, or at least reviewers. There is no listed repository that I could see linked in any of the materials or supplemental information that would let me see the raw data from this study, and the results are described at an extremely coarse level of detail. While the authors claim in the data availability statement that the data are available in the supporting information files, no information on Study 1 is found in either appendix, and no other files were in what I, as a reviewer, was able to download. At a minimum, I would have expected a table showing the individual items in each cluster in the mechanism condition (if not in each condition), since the identification of the clusters as "mechanical", "chemical", and "electromagnetic" is not currently supported in any way other than the author's assertion and a handful of examples from each cluster.

However, Study 1 may also be simply redundant, because each experiment includes its own stimulus validation experiment. Right now, with so little information about Study 1 available, I read the categories as largely arbitrarily defined, but that's not necessarily a problem given 1) the validation in each experiment and 2) the consistency of the results. While it would be nice to have a more complete version of Study 1 to give stronger external validation of the categories, the complexity of the analysis, and the underlying degrees of experimenter freedom about, for example, the cutoffs, mean that the paper may be most improved by simply omitting it and treat the three categories as being defined a priori. Since the stimuli of Studies 2-5 are not in fact based of the body of stimuli used in Study 1, Study 1 doesn't even provide direct evidence for these categories as they apply in the other experiments regardless.

That said, studies 2-5 are excellent tests of the idea that people cluster mechanisms by domain. Study 5 in particular provides an elegant control for any concerns related to the individual stimuli. I believe that these studies fit all the criteria for publication in PLoS ONE, and either providing much more information about Study 1 or dropping it altogether will yield a paper that is suitable for publication.

Minor things:

Figs. 1 and 2 have low image resolution, though I expect this will be fixed during production.

Line 341: The base rate experiment is Study 3, not Study 4.

Line 453: "previous chapter" previous experiment?

6. PLOS authors have the option to publish the peer review history of their article (what does this mean?). If published, this will include your full peer review and any attached files.

Reviewer #1: **Yes: **Sam Johnson

Reviewer #2: **Yes: **Simon Stephan

Reviewer #3: **Yes: **Jonathan F. Kominsky

---

## [Author Response · Author response to Decision Letter 0]

24 Sep 2021

Response letter has been uploaded

---

## [Decision Letter · Decision Letter 1]

17 Jan 2022

PONE-D-21-08341R1Causal reasoning without mechanismPLOS ONE

Dear Dr. Dündar-Coecke,

thank you for submitting the revision of your manuscript to PLOS ONE. After careful consideration, we feel that it has merit but does not fully meet PLOS ONE’s publication criteria as it currently stands. Therefore, we invite you to submit a revised version of the manuscript that carefully addresses the (technical) points raised by Reviewer 2, the thorough duicussion of which you will find below.

Please submit your revised manuscript within six months from this date as afterwards any revision has to be considered a new submission. If you will need more time than this to complete your revisions, please reply to this message or contact the journal office at plosone@plos.org. Please include the following items when submitting your revised manuscript:A rebuttal letter that responds to each point raised by the academic editor and reviewer(s). You should upload this letter as a separate file labeled 'Response to Reviewers'.A marked-up copy of your manuscript that highlights changes made to the original version. You should upload this as a separate file labeled 'Revised Manuscript with Track Changes'.An unmarked version of your revised paper without tracked changes. You should upload this as a separate file labeled 'Manuscript'.

We look forward to receiving your revised manuscript.

Thank you for submitting your reseach to PLOS ONE.

Kind regards and stay safe and healthy in 2022,

Sasha

Alexander N. 'Sasha' Sokolov, Ph.D.

Academic Editor

PLOS ONE

Reviewers' comments:

Reviewer's Responses to Questions

**Comments to the Author**

1. If the authors have adequately addressed your comments raised in a previous round of review and you feel that this manuscript is now acceptable for publication, you may indicate that here to bypass the “Comments to the Author” section, enter your conflict of interest statement in the “Confidential to Editor” section, and submit your "Accept" recommendation.

Reviewer #1: All comments have been addressed

Reviewer #2: (No Response)

Reviewer #3: All comments have been addressed

2. Is the manuscript technically sound, and do the data support the conclusions?

Reviewer #1: Yes

Reviewer #2: Partly

Reviewer #3: Yes

3. Has the statistical analysis been performed appropriately and rigorously? 

Reviewer #1: Yes

Reviewer #2: Yes

Reviewer #3: Yes

4. Have the authors made all data underlying the findings in their manuscript fully available?

Reviewer #1: Yes

Reviewer #2: Yes

Reviewer #3: No

5. Is the manuscript presented in an intelligible fashion and written in standard English?

Reviewer #1: Yes

Reviewer #2: Yes

Reviewer #3: Yes

6. Review Comments to the Author

Reviewer #1: I am pleased to report that the authors have done a good job of addressing my comments. The issue of “domains” in particular was much clearer in the revision – although this is a fairly small change in terms of prose, I think it makes the theoretical picture much clearer. Overall, I believe this article represents an advance in our understanding of causal inference and I am happy to support publication. Thanks for the opportunity to be one of your first readers!

Sam Johnson

Reviewer #2: Review of “Causal reasoning without mechanism” #Revision 1 by Dündar-Coecke, Goldin, and Sloman

I’d like to apologize for the delayed submission of my review. I was Reviewer #2 in the first review round and now got the chance to read the revised manuscript.

My overall impression hasn’t changed. I think that this is very interesting work with the potential to make a relevant contribution to the literature on causal cognition.

I also think that the authors did an overall great job (though see below) in addressing the concerns and suggestions provided by the three reviewers. My two main former concerns were:

(1) “However, I think that the writing requires some polishing; my main point here is that I think that the paragraphs are often to loosely connected.

I think that the authors have addressed this concern very well.

(2) I also think that DGS need to address the causal strength argument, although I unfortunately can’t suggest a specific experiment right now that would address this issue straightforwardly.”

By contrast, I think that this second of my concerns has still not been resolved satisfactorily. My feelings that many of the thoughts the authors have provided in their reply (in the response letter) to this concern would be worth including in the discussion of the results, or even the general discussion. The authors in their reply wrote “The causal strength of each cause for specific effects is higher when the causes match the domain of the effect. This is a possibility in principle. Our claim is that the reason for greater causal strength is because the cause and effect match domain. Perhaps each of our items suffers from a different confound, but the one invariant across items is that causal strength is higher when domains match.”. If domain-matching and causal strength were to be prima facie confounded, shouldn’t this interesting observation be presented and discussed? The authors continued “We take this as pretty good evidence that the reason that one cause has greater causal strength for one effect and not the other is because it shares a domain with the former.”. I tend to agree with this point, but doesn’t it ultimately mean that there is a confounding explanation here? Do reasoners primarily “think” of domain matching or are they relying on knowledge about strength? What is the psychological primary factor here that drives their decisions? I think this point should be discussed (if not tested, but this could maybe be left open for follow-up studies). However, I also feel inclined to mention that the two other reviewers did not seem to share this concern, so maybe I’m being too strict here.

A remark on the sample size rationale: the authors now write “Using the G-Power software, minimum power to detect effect sizes in ANOVAs was .80 – a generally accepted level of power.” What is this supposed to mean? A given (or in this case aspired) level of test power for a significance test is relative to an assumed (standardized) effect size. So what readers need to know here is for which effect size estimate (or assumed effect size) the authors aimed to achieve 80 percent test power. Also, why talk about ANOVAs here? Unless I’m missing something here, the tests testing the domain matching theory aren’t ANOVAs. The authors test proportions here. There’s an ANOVA to test age effects of domain matching, but this question was rather a side than a main issue. My suggestion is that the authors either revise this sentence or, in case they actually didn’t base their sample sizes on effect size assumptions, leave it out completely.

Minor things:

Line 46: “[…] knowledge that in fact we do.” � “than”

Line 128: the paragraph starts with “Schultz’s study also shows […]” � unless I missed it, this is the first mentioning of that study, so it is weird to start a paragraph about it as if the readers already know what Shultz et al. did.

Line 543: “[…] cause even if when those […]” � either “if” or “when”.

Signed (24 Nov 2021),

Simon Stephan

Reviewer #3: The figures and added information for Experiment 1 is extremely helpful and justifies its inclusion in the paper. The full data for Experiment 1 are still not accessible at least to reviewers, but there is enough information now to make sense of the results. The other changes are all good as well. I am happy to recommend this paper for publication in PLoS ONE.

7. PLOS authors have the option to publish the peer review history of their article (what does this mean?). If published, this will include your full peer review and any attached files.

Reviewer #1: **Yes: **Sam Johnson

Reviewer #2: **Yes: **Simon Stephan

Reviewer #3: No

---

## [Author Response · Author response to Decision Letter 1]

29 Jan 2022

The response letter has been uploaded as a separate file.

---

## [Decision Letter · Decision Letter 2]

10 Feb 2022

PONE-D-21-08341R2Causal reasoning without mechanismPLOS ONE

Dear Dr. Dündar-Coecke,

Thank you for submitting your revised manuscript to PLOS ONE.  After careful consideration by several experts in the field and myself, we feel it represents a nice piece of research, has been substantially improved, and can be recommended for publication conditional upon fulfilling PLOS ONE’s publication criteria as to the statistical reports, and considering minor suggestions of Reviewer 2 below.  Therefore, we invite you to submit a revised version of the manuscript that addresses the following:  Please specify in the manuscript whether or not (1) tests for normality of data sets have been run throughout (if yes, please identify which) and the data processing accomplished accordingly, and (2) corrections for multiple comparisons have been employed as necessary.  Should some concerns arise please explain why these procedures have been deemed not applicable.  Finally, please consider the minor points of Reviewer 2 as listed below.

Please submit your revised manuscript within six months from this date as at a later time point, any revision will have to be considered a new submission.  If you will need more time than this to complete your revisions, please reply to this message or contact the journal office at plosone@plos.org. Please include the following items when submitting your revised manuscript:A rebuttal letter that responds to each point raised by the academic editor and reviewer(s). You should upload this letter as a separate file labeled 'Response to Reviewers'.A marked-up copy of your manuscript that highlights changes made to the original version. You should upload this as a separate file labeled 'Revised Manuscript with Track Changes'.An unmarked version of your revised paper without tracked changes. You should upload this as a separate file labeled 'Manuscript'.

We look forward to receiving your revised manuscript. 

Thank you for considering PLOS ONE for reporting your research.

Kind regards,

Sasha

Alexander N. 'Sasha' Sokolov, Ph.D.

Academic Editor

PLOS ONE

Journal Requirements:

Reviewers' comments:

Reviewer's Responses to Questions

**Comments to the Author**

1. If the authors have adequately addressed your comments raised in a previous round of review and you feel that this manuscript is now acceptable for publication, you may indicate that here to bypass the “Comments to the Author” section, enter your conflict of interest statement in the “Confidential to Editor” section, and submit your "Accept" recommendation.

Reviewer #2: All comments have been addressed

2. Is the manuscript technically sound, and do the data support the conclusions?

Reviewer #2: Yes

3. Has the statistical analysis been performed appropriately and rigorously? 

Reviewer #2: Yes

4. Have the authors made all data underlying the findings in their manuscript fully available?

Reviewer #2: Yes

5. Is the manuscript presented in an intelligible fashion and written in standard English?

Reviewer #2: Yes

6. Review Comments to the Author

Reviewer #2: Review of Revision #2 of “Cauasal reasoning without mechanism” by Dündar-Coecke, Goldin, and Sloman

I was Reviewer #2 and now got the chance to read the authors’ second revision of this interesting paper. The concern that I raised in my first and also in my second review was:

“(2) I also think that DGS need to address the causal strength argument, although I unfortunately can’t suggest a specific experiment right now that would address this issue straightforwardly.”

The authors now added a longer paragraph in which they discuss this point in greater detail and I think that this enough. I don’t have any more substantial concerns. This is an interesting paper that many will like to read.

Minor comments:

Line 636: The Chi-Sq. test statistic is written as “X2”

It was quite a pain that all the figures were placed at the end of the manuscript and not in the text. Having to jump back and forth while reading was not a very pleasant experience.

Thanks for giving me the opportunity to read this interesting work as one of the first.

Signed (February, 09, 2022),

Simon Stephan

7. PLOS authors have the option to publish the peer review history of their article (what does this mean?). If published, this will include your full peer review and any attached files.

Reviewer #2: **Yes: **Simon Stephan

---

## [Author Response · Author response to Decision Letter 2]

17 Feb 2022

Minor revisions were explained above.

---

## [Editor Report · Decision Letter 3]

26 Apr 2022

Causal reasoning without mechanism

PONE-D-21-08341R3

Dear Dr. Dündar-Coecke,

thank you for the revision of your above manuscript.  We’re happy to inform you that your revised manuscript has been judged scientifically suitable for publication and will be formally accepted for publication once it meets all outstanding technical requirements.

Thank you for choosing PLOS ONE for reporting your research.

Kind regards,

Sasha

Alexander N. 'sasha' Sokolov, Ph.D.

Academic Editor

PLOS ONE
---

## [Editor Report · Acceptance letter]

28 Apr 2022

PONE-D-21-08341R3 

Causal reasoning without mechanism 

Dear Dr. Dündar-Coecke:

I'm pleased to inform you that your manuscript has been deemed suitable for publication in PLOS ONE. Congratulations! Your manuscript is now with our production department. 

Kind regards, 

on behalf of

Dr. Alexander N. Sokolov 

Academic Editor

PLOS ONE